# QUADRATURE-BASED FEATURES FOR KERNEL APPROXIMATION

## ABSTRACT

We consider the problem of improving kernel approximation via randomized feature maps. These maps arise as Monte Carlo approximation to integral representations of kernel functions and scale up kernel methods for larger datasets. We propose to use more efficient numerical integration technique to obtain better estimates of the integrals compared to the state-of-the-art methods. Our approach allows the use of information about the integrand to enhance approximation and facilitates fast computations. We derive the convergence behavior and conduct an extensive empirical study that supports our hypothesis.

## 1 INTRODUCTION

Kernel methods proved to be an efficient technique in numerous real-world problems. The core idea of kernel methods is the kernel trick – compute an inner product in a high-dimensional (or even infinite-dimensional) feature space by means of a kernel function $k$:

$$k(\mathbf{x}, \mathbf{y}) = \langle \psi(\mathbf{x}), \psi(\mathbf{y}) \rangle, \tag{1}$$

where $\psi : \mathcal{X} \to \mathcal{F}$ is a non-linear feature map transporting elements of input space $\mathcal{X}$ into a feature space $\mathcal{F}$. It is a common knowledge that kernel methods incur space and time complexity infeasible to be used with large-scale datasets directly. For example, kernel regression has $\mathcal{O}(N^3 + Nd^2)$ training time, $\mathcal{O}(N^2)$ memory, $\mathcal{O}(Nd)$ prediction time complexity for $N$ data points in original $d$-dimensional space $\mathcal{X}$. One of the most successful techniques to handle this problem (Rahimi & Recht (2008)) introduces a low-dimensional randomized approximation to feature maps:

$$k(\mathbf{x}, \mathbf{y}) \approx \hat{\boldsymbol{\Psi}}(\mathbf{x})^\top \hat{\boldsymbol{\Psi}}(\mathbf{y}). \tag{2}$$

This is essentially carried out by using Monte-Carlo sampling to approximate scalar product in (1). A randomized $D$-dimensional mapping $\hat{\boldsymbol{\Psi}}(\cdot)$ applied to the original data input allows employing standard linear methods, i.e. reverting the kernel trick. In doing so one reduces the complexity to that of linear methods, e.g. $D$-dimensional approximation admits $\mathcal{O}(ND^2)$ training time, $\mathcal{O}(ND)$ memory and $\mathcal{O}(N)$ prediction time.

It is well known that as $D \to \infty$, the inner product in (2) converges to exact kernel $k(\mathbf{x}, \mathbf{y})$. Recent research (Yang et al. (2014); Felix et al. (2016); Choromanski & Sindhwani (2016)) aims to improve the convergence of approximation so that a smaller $D$ can be used to obtain the same quality of approximation.

This paper considers kernels that allow the following integral representation

$$k(\mathbf{x}, \mathbf{y}) = \mathbb{E}_{q(\mathbf{w})} g_{\mathbf{xy}}(\mathbf{w}) \approx \mathbb{E}_{p(\mathbf{w})} f_{\mathbf{xy}}(\mathbf{w}) = I(f_{\mathbf{xy}}), \quad p(\mathbf{w}) = \frac{1}{(2\pi)^{d/2}} e^{-\frac{\|\mathbf{w}\|^2}{2}}, \tag{3}$$

where $q(\mathbf{w})$ is a density associated with a kernel, e.g. the popular Gaussian kernel has $q(\mathbf{w}) = p(\mathbf{w})$, so the exact equality holds with $g_{\mathbf{xy}}(\mathbf{w}) = f_{\mathbf{xy}}(\mathbf{w}) = \phi(\mathbf{w}^\top \mathbf{x})^\top \phi(\mathbf{w}^\top \mathbf{y})$, where $\phi(\cdot) = [\cos(\cdot), \sin(\cdot)]^\top$.

The class of kernels admitting the form in (3) covers shift-invariant kernels (e.g. radial basis function (RBF) kernels) and Pointwise Nonlinear Gaussian (PNG) kernels. They are widely used in practice and have interesting connections with neural networks (Cho & Saul (2009), Williams (1997)).

The main challenge for the construction of low-dimensional feature maps is the approximation of the expectation in (3) which is $d$-dimensional integral with Gaussian weight. While standard Monte-Carlo rule is easy to implement, there are better quadrature rules for such kind of integrals. For example, Yang et al. (2014) apply quasi-Monte Carlo (QMC) rules and obtain better quality kernel matrix approximations compared to random Fourier features of Rahimi & Recht (2008).

Unlike other research studies we refrain from using simple Monte Carlo estimate of the integral, instead, we propose to use specific quadrature rules. We now list our contributions:

1. We propose to use advanced quadrature rules to improve kernel approximation accuracy. We also provide an analytical estimate of the error for the used quadrature rules.

2. We note that for kernels with specific integrand $f_{\mathbf{xy}}(\mathbf{w})$ in (3) one can improve on its properties. For example, for kernels with even function $f_{\mathbf{xy}}(\mathbf{w})$ we derive the reduced quadrature rule which gives twice smaller embedded dimension $D$ with the same accuracy. This applies, for example, to any RBF kernel.

3. We use structured orthogonal matrices (so-called *butterfly matrices*) when designing quadrature rule that allow fast matrix by vector multiplications. As a result, we speed up the approximation of the kernel function and reduce memory requirements.

4. We demonstrate our approach on a set of regression and classification problems. Empirical results show that the proposed approach has a better quality of approximation of kernel function as well as better quality of classification and regression when using different kernels.

## 2 QUADRATURE-BASED RANDOM FEATURES

We start with rewriting the expectation in equation (3) as integral of $f_{\mathbf{xy}}$ with respect to $p(\mathbf{w})$:

$$I(f_{\mathbf{xy}}) = (2\pi)^{-\frac{d}{2}} \int_{-\infty}^{\infty} \cdots \int_{-\infty}^{\infty} e^{-\frac{\mathbf{w}^\top \mathbf{w}}{2}} f_{\mathbf{xy}}(\mathbf{w}) d\mathbf{w}.$$

Integration can be performed by means of quadrature rules. The rules usually take a form of interpolating function that is easy to integrate. Given such a rule, one may sample points from the domain of integration and calculate the value of the rule at these points. Then, the sample average of the rule values would yield the approximation of the integral.

We use the average of sampled quadrature rules developed by Genz & Monahan (1998) to yield unbiased estimates of $I(f_{\mathbf{xy}})$. A change of coordinates is the first step to facilitate stochastic spherical-radial rules. Now, let $\mathbf{w} = r\mathbf{z}$, with $\mathbf{z}^\top \mathbf{z} = 1$, so that $\mathbf{w}^\top \mathbf{w} = r^2$ for $r \in [0, \infty]$, leaving us with

$$I(f_{\mathbf{xy}}) = (2\pi)^{-\frac{d}{2}} \int_{U_d} \int_0^{\infty} e^{-\frac{r^2}{2}} r^{d-1} f_{\mathbf{xy}}(r\mathbf{z}) dr d\mathbf{z} = \frac{(2\pi)^{-\frac{d}{2}}}{2} \int_{U_d} \int_{-\infty}^{\infty} e^{-\frac{r^2}{2}} |r|^{d-1} f_{\mathbf{xy}}(r\mathbf{z}) dr d\mathbf{z}, \quad (4)$$

where $U_d = \{\mathbf{z} : \mathbf{z}^\top \mathbf{z} = 1, \mathbf{z} \in \mathbb{R}^d\}$. As mentioned earlier we are going to use a combination of radial $R$ and spherical $S$ rules. We now describe the logic behind the used quadratures.

**Stochastic radial rules.** Stochastic radial rule $R(h)$ of degree $2l + 1$ has the form of weighted symmetric sums:

$$R(h) = \sum_{i=0}^{l} w_i \frac{h(\rho_i) + h(-\rho_i)}{2},$$

where $h$ is an integrand in infinite range integral $T(h) = \int_{-\infty}^{\infty} e^{-\frac{r^2}{2}} |r|^{d-1} h(r) dr$. To get an unbiased estimate for $T(h)$, points $\rho_i$ are sampled from specific distributions which depend on the degree of the rule. Weights $w_i$ are derived so that $R$ has a polynomial degree $2l+1$, i.e. is exact for integrands $h(r) = r^p$ with $p = 0, 1, \ldots, 2l + 1$. For radial rules of degree three $R^3$ the point $\rho_0 = 0$, while $\rho_1 \sim \chi(d + 2)$ follows Chi-distribution with $d + 2$ degrees of freedom. Higher degrees require samples from more complex distributions which are hard to sample from.

**Stochastic spherical rules.** Spherical rule $S(s)$ approximates an integral of a function $s(\mathbf{z})$ over the surface of unit $d$-sphere $U_d$ and takes the following form:

$$S(s) = \sum_{j=1}^{p} \widetilde{w}_j s(\mathbf{z}_j),$$

where $\mathbf{z}_j$ are points on $U_d$, i.e. $\mathbf{z}^\top \mathbf{z} = 1$. If we set weight $\widetilde{w}_j = \frac{|U_d|}{2(d+1)}$ and sum function $s$ values at original and reflected vertices $\mathbf{v}_j$ of randomly rotated $d$-simplex $\mathbf{V}$, we will end up with a degree three rule:

$$S_\mathbf{Q}^3(f_{\mathbf{xy}}(\rho\mathbf{Qz})) = \frac{|U_d|}{2(d+1)} \sum_{j=1}^{d+1} \left[ f_{\mathbf{xy}}(-\mathbf{Qv}_j) + f_{\mathbf{xy}}(\mathbf{Qv}_j) \right],$$

where $\mathbf{v}_j$ is the $j$'th vertex of $d$-simplex $\mathbf{V}$ with vertices on $U_d$ and $\mathbf{Q}$ is a random $d \times d$ orthogonal matrix. We justify the choice of the degree in Appendix A.

Since the value of the integral is approximated as the sample average, the key to unbiased estimate is proper randomization. In this case, randomization is attained through the matrix $\mathbf{Q}$. It is crucial to generate uniformly random orthogonal matrices to achieve an unbiased estimate for spherical surface integrals. We consider various designs of such matrices further in Section 3.

**Stochastic spherical-radial rules.** Meanwhile, combining foregoing rules results in stochastic spherical-radial rule of degree three:

$$SR_{\mathbf{Q},\rho}^{3,3}(f_{\mathbf{xy}}) = \left(1 - \frac{d}{\rho^2}\right) f_{\mathbf{xy}}(\mathbf{0}) + \frac{d}{d+1} \sum_{j=1}^{d+1} \left[ \frac{f_{\mathbf{xy}}(-\rho\mathbf{Qv}_j) + f_{\mathbf{xy}}(\rho\mathbf{Qv}_j)}{2\rho^2} \right], \qquad (5)$$

which we finally apply to the approximation of (4) by averaging the samples of $SR_{\mathbf{Q},\rho}^{3,3}$:

$$I(f) = \mathbb{E}_{\mathbf{Q},\rho}[SR_{\mathbf{Q},\rho}^{3,3}(f_{\mathbf{xy}})] \approx \hat{I}(f_{\mathbf{xy}}) = \frac{1}{n} \sum_{i=1}^{n} SR_{\mathbf{Q}_i,\rho_i}^{3,3}(f_{\mathbf{xy}}), \qquad (6)$$

where $n$ is the number of sampled $SR$ rules. Speaking in terms of approximate feature maps, the new feature dimension $D$ in case of quadrature based approximation equals $2n(d+1)$ as we sample $n$ rules and evaluate each of them at $2(d+1)$ points. Surprisingly, empirical results (see Section 5) show that even a small number of rule samples $n$ provides accurate approximations.

**Properties of the integrand.** We also note here that for specific functions $f_{\mathbf{xy}}(\mathbf{w})$ we can derive better versions of $SR$ rule by taking on advantage of the knowledge about the integrand. For example, the Gaussian kernel has $f_{\mathbf{xy}}(\mathbf{w}) = \cos(\mathbf{w}^\top(\mathbf{x} - \mathbf{y}))$. Note that $f$ is even, so we can discard an excessive term in the summation in (5), since $f(\mathbf{w}) = f(-\mathbf{w})$, i.e $SR^{3,3}$ rule reduces to

$$SR_{\mathbf{Q},\rho}^{3,3}(f_{\mathbf{xy}}) = \left(1 - \frac{d}{\rho^2}\right) f_{\mathbf{xy}}(\mathbf{0}) + \frac{d}{d+1} \sum_{j=1}^{d+1} \frac{f_{\mathbf{xy}}(\rho\mathbf{Qv}_j)}{\rho^2}. \qquad (7)$$

**Variance of the error.** We contribute the variance estimation for the stochastic spherical-radial rules when applied to kernel function. To the best of our knowledge, it has not been done before. In case of kernel functions the integrand $f_{\mathbf{xy}}$ can be represented as

$$f_{\mathbf{xy}}(\mathbf{w}) = \phi(\mathbf{w}^\top\mathbf{x})^\top \phi(\mathbf{w}^\top\mathbf{y}) = g(z_1, z_2), \quad z_1 = \mathbf{w}^\top\mathbf{x}, \quad z_2 = \mathbf{w}^\top\mathbf{y}, \qquad (8)$$

where $z_1, z_2$ are scalar values. Using this representation of kernel function and its Taylor expansion we can obtain the following proposition (see Appendix B for detailed derivation of the result):

**Proposition 2.1.** The quadrature rule (6) is an unbiased estimate of integral of any integrable function $f$. If function $f$ can be represented in the form (8), i.e. $f(\mathbf{w}) = g(z_1, z_2)$, $z_1 = \mathbf{w}^\top\mathbf{x}$, $z_2 = \mathbf{w}^\top\mathbf{y}$ for some $\mathbf{x}, \mathbf{y} \in \mathbb{R}^d$, all 4-th order partial derivatives of $g$ are bounded and $D = 2n(d+1)$ is the number of generated features, then

$$\mathbb{V}[\hat{I}(f)] \leq \frac{2.66 M_1^2 L^8}{nd} + \frac{212 M_1 M_2 L^6}{nd^3} + \frac{(d+95)M_2^2 L^4}{4nd^2(d-2)},$$

where $M_1 = \max\left\{\sup_z \left|\frac{\partial^4 g}{\partial z_1^4}\right|, \sup_z \left|\frac{\partial^4 g}{\partial z_2^4}\right|, \sup_z \left|\frac{\partial^4 g}{\partial z_1^2 \partial z_2^2}\right|\right\}, M_2 = \max_{j=0,1,2}\left\{\left|\frac{\partial^2 g}{\partial z_1^j \partial z_2^{2-j}}(0,0)\right|\right\}$ and $L = \max\{\|\mathbf{x}\|, \|\mathbf{y}\|\}$.

Constants $M_1, M_2$ in the proposition are upper bounds on the derivatives of function $g$ and don't depend on the data set, while $L$ plays the role of the scale of inputs. The proposition implies that the error of approximation is proportional to $L$ – the less the scale, the better the accuracy (see Figure 1). However, scaling input vectors is equivalent to changing the parameters of the kernel function. For example, decreasing the norm of input variables for RBF kernel is equivalent to increasing the kernel width $\sigma$. Therefore, the wide RBF kernels are approximated better than the narrow ones. This result also gives us the rate of convergence $O(1/nd)$ for the quadrature rule.

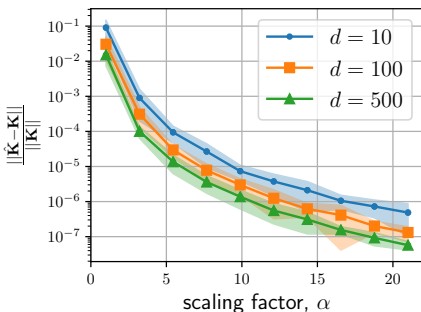

Figure 1: Relative error of approximation of kernel matrix with $95\%$ confidence interval depending on the scaling factor, Gaussian kernel was used. In this experiment for each scaling factor $\alpha$ we construct approximate kernel matrix $\widehat{\mathbf{K}}$ and the exact kernel matrix $\mathbf{K}$ using scaled input vectors $\tilde{\mathbf{x}} = \mathbf{x}/\alpha$. To plot the confidence interval we run each experiment 10 times each time generating new weights. Experiment was conducted for 3 different input dimensions: $d = 10, 100, 500$.

The quadrature rule (5) grants us some freedom in the choice of random orthogonal matrix $\mathbf{Q}$. The next section discusses such matrices and suggests *butterfly matrices* for fast matrix by vector multiplication as the $SR^{3,3}$ rule implementation involves multiplication of the matrix $\mathbf{QV}$ by the data vector $\mathbf{x}$.

## 3 GENERATING UNIFORMLY RANDOM ORTHOGONAL MATRICES

Previously described stochastic spherical-radial rules require a random orthogonal matrix $\mathbf{Q}$ (see equation (5)). If $\mathbf{Q}$ follows Haar distribution on the set of all matrices in the orthogonal group $\mathrm{O}(d)$ in dimension $d$, then the averages of spherical rules $S_{\mathbf{Q}_i}^3(s)$ provide unbiased degree three estimates for integrals over unit sphere. Essentially, Haar distribution means that all orthogonal matrices in the group are equiprobable, i.e. uniformly random. Methods for sampling such matrices vary in their complexity of generation and multiplication.

Techniques based on QR decomposition (Mezzadri (2006)) have complexity cubic in $d$, and the resulting matrix does not allow fast matrix by vector multiplications. Another set of methods is based on a sequence of reflectors (Stewart (1980)) or rotators (Anderson et al. (1987)). The complexity is better (quadratic in $d$), however the resulting matrix is unstructured and, thus, implicates no fast matrix by vector multiplication. In Choromanski et al. (2017) random orthogonal matrices are considered. They are constructed as a product of random diagonal matrices and Hadamard matrices and therefore enable fast matrix by vector products. Unfortunately, they are not guaranteed to follow the Haar distribution.

To satisfy both our requirements, i.e low computational/space complexity and generation of Haar distributed orthogonal matrices, we propose to use so-called *butterfly matrices*.

**Butterfly matrices.** The method from Genz (1998) generates Haar distributed random orthogonal matrix $\mathbf{B}$. As it happens to be a product of butterfly structured factors, a matrix of this type conveniently possesses the property of fast multiplication. For $d = 4$ an example of butterfly orthogonal

matrix is

$$\mathbf{B}^{(4)} = \begin{bmatrix} c_1 & -s_1 & 0 & 0 \\ s_1 & c_1 & 0 & 0 \\ 0 & 0 & c_3 & -s_3 \\ 0 & 0 & s_3 & c_3 \end{bmatrix} \begin{bmatrix} c_2 & 0 & -s_2 & 0 \\ 0 & c_2 & 0 & -s_2 \\ s_2 & 0 & c_2 & 0 \\ 0 & s_2 & 0 & c_2 \end{bmatrix} = \begin{bmatrix} c_1c_2 & -s_1c_2 & -c_1s_2 & s_1s_2 \\ s_1c_2 & c_1c_2 & -s_1s_2 & -c_1s_2 \\ c_3s_2 & -s_3s_2 & c_3c_2 & -s_3c_2 \\ s_3s_2 & c_3s_2 & s_3c_2 & c_3c_2 \end{bmatrix}.$$

**Definition 3.1.** Let $c_i = \cos\theta_i$, $s_i = \sin\theta_i$ for $i = 1, \ldots, d-1$ be given. Assume $d = 2^k$ with $k > 0$. Then an orthogonal matrix $\mathbf{B}^{(d)} \in \mathbb{R}^{d \times d}$ is defined recursively as follows

$$\mathbf{B}^{(2d)} = \begin{bmatrix} \mathbf{B}^{(d)}c_d & -\mathbf{B}^{(d)}s_d \\ \hat{\mathbf{B}}^{(d)}s_d & \hat{\mathbf{B}}^{(d)}c_d \end{bmatrix}, \quad \mathbf{B}^{(1)} = 1,$$

where $\hat{\mathbf{B}}^{(d)}$ is the same as $\mathbf{B}^{(d)}$ with indexes $i$ shifted by $d$, e.g.

$$\mathbf{B}^{(2)} = \begin{bmatrix} c_1 & -s_1 \\ s_1 & c_1 \end{bmatrix}, \quad \hat{\mathbf{B}}^{(2)} = \begin{bmatrix} c_3 & -s_3 \\ s_3 & c_3 \end{bmatrix}.$$

Matrix $\mathbf{B}^{(d)}$ by vector product has computational complexity $O(d \log d)$ since $\mathbf{B}^{(d)}$ has $\lceil \log d \rceil$ factors and each factor requires $O(d)$ operations. Another advantage is space complexity: $\mathbf{B}^{(d)}$ is fully determined by $d-1$ angles $\theta_i$, yielding $O(d)$ memory complexity.

One can easily define butterfly matrix $\mathbf{B}^{(d)}$ for the cases when $d$ is not a power of two (see Appendix C.1 for details). The randomization is based on the sampling of angles $\theta$ and we discuss it in Appendix C.2. The method that uses butterfly orthogonal matrices is denoted $\mathbf{B}$ in the experiments section.

## 4 KERNELS

This section gives examples on how quadrature rules can be applied to a number of kernels.

### 4.1 GAUSSIAN KERNEL

Radial basis function (RBF) kernels are popular kernels widely used in kernel methods. Gaussian kernel is a widely exploited RBF kernel and has the following form:

$$k(\mathbf{x}, \mathbf{y}) = \exp\left(-\frac{\|\mathbf{x} - \mathbf{y}\|^2}{2\sigma^2}\right).$$

In this case the integral representation has $\phi(\mathbf{w}^\top\mathbf{x}) = [\cos(\mathbf{w}^\top\mathbf{x}), \sin(\mathbf{w}^\top\mathbf{x})]^\top$. Since $f_{\mathbf{xy}}(0) = 1$, $SR^{3,3}$ rule for Gaussian kernel has the form ($\sigma$ appears due to scaling):

$$SR^{3,3}_{\mathbf{Q},\rho}(f_{\mathbf{xy}}) = \left(1 - \frac{d}{\rho^2}\right) + \frac{d}{d+1}\sum_{j=1}^{d+1}\frac{f_{\mathbf{xy}}\left(\frac{\rho\mathbf{Q}\mathbf{v}_j}{\sigma}\right)}{\rho^2},$$

### 4.2 ARC-COSINE KERNELS

Arc-cosine kernels were originally introduced by Cho & Saul (2009) upon studying the connections between deep learning and kernel methods. The integral representation of the $b^{th}$-order arc-cosine kernel is

$$k_b(\mathbf{x}, \mathbf{y}) = 2\int_{\mathbb{R}^d} \phi_b(\mathbf{w}^\top\mathbf{x})\phi_b(\mathbf{w}^\top\mathbf{y})p(\mathbf{w})d\mathbf{w},$$

where $\phi_b(\mathbf{w}^\top\mathbf{x}) = \Theta(\mathbf{w}^\top\mathbf{x})(\mathbf{w}^\top\mathbf{x})^b$, $\Theta(\cdot)$ is the Heaviside function and $p$ is the density of the standard Gaussian distribution. Such kernels can be seen as an inner product between the representation produced by infinitely wide single layer neural network with random Gaussian weights. They have closed form expression in terms of the angle $\theta = \cos^{-1}\left(\frac{\mathbf{x}^\top\mathbf{y}}{\|\mathbf{x}\|\|\mathbf{y}\|}\right)$ between $\mathbf{x}$ and $\mathbf{y}$.

$0^{th}$-order arc-cosine kernel is given by $k_0(\mathbf{x}, \mathbf{y}) = 1 - \frac{\theta}{\pi}$, $1^{st}$-order kernel is given by $k_1(\mathbf{x}, \mathbf{y}) = \frac{\|\mathbf{x}\|\|\mathbf{y}\|}{\pi}(\sin\theta + (\pi - \theta)\cos\theta)$.

Table 1: Space and time complexity for different kernel approximation algorithms.

| Method | Space | Time |
|---|---|---|
| ORF | $\mathcal{O}(Dd)$ | $\mathcal{O}(Dd)$ |
| QMC | $\mathcal{O}(Dd)$ | $\mathcal{O}(Dd)$ |
| ROM | $\mathcal{O}(d)$ | $\mathcal{O}(d \log d)$ |
| **Quadrature based** | $\mathcal{O}(d)$ | $\mathcal{O}(d \log d)$ |

Let $\phi_0(\mathbf{w}^\top \mathbf{x}) = \Theta(\mathbf{w}^\top \mathbf{x})$ and $\phi_1(\mathbf{w}^\top \mathbf{x}) = \max(0, \mathbf{w}^\top \mathbf{x})$, then we can rewrite the integral representation as follows: $k_b(\mathbf{x}, \mathbf{y}) = 2 \int_{\mathbb{R}^d} \phi_b(\mathbf{w}^\top \mathbf{x}) \phi_b(\mathbf{w}^\top \mathbf{y}) p(\mathbf{w}) d\mathbf{w} \approx \frac{2}{n} \sum_{i=1}^{n} SR^{3,3}_{\mathbf{Q}_i, \boldsymbol{\rho}_i}$. For arc-cosine kernel of order 0 the value of the function $\phi_0(0) = \Theta(0) = 0.5$ results in

$$SR^{3,3}_{\mathbf{Q},\rho}(f_{\mathbf{xy}}) = 0.25 \left(1 - \frac{d}{\rho^2}\right) + \frac{d}{d+1} \sum_{j=1}^{d+1} \frac{f_{\mathbf{xy}}(\rho \mathbf{Q} \mathbf{v}_j) + f_{\mathbf{xy}}(-\rho \mathbf{Q} \mathbf{v}_j)}{2\rho^2}.$$

In the case of arc-cosine kernel of order 1, the value of $\phi_1(0)$ is 0 and the $SR^{3,3}$ rule reduces to

$$SR^{3,3}_{\mathbf{Q},\rho}(f_{\mathbf{xy}}) = \frac{d}{d+1} \sum_{j=1}^{d+1} \frac{f_{\mathbf{xy}}(|\rho \mathbf{Q} \mathbf{v}_j|)}{2\rho^2}.$$

### 4.3 EXPLICIT MAPPING

The explicit mapping can be written as follows:

$$\psi(\mathbf{x}) = \begin{bmatrix} a_0 f_{\mathbf{xy}}(0) & a_1 f_{\mathbf{xy}}(\mathbf{w}_1^\top \mathbf{x}) & \dots & a_D f_{\mathbf{xy}}(\mathbf{w}_D^\top \mathbf{x}) \end{bmatrix}$$

where $a_0 = \sqrt{1 - \frac{d}{\rho^2}}$[1], for $i = 1, \dots, D$   $a_i = \frac{1}{\rho}\sqrt{\frac{d}{2(d+1)}}$, $\mathbf{w}_i$ is the row in matrix $\mathbf{W}$. The matrix $\mathbf{W} = \rho \begin{bmatrix} (\mathbf{Q}\mathbf{V})^\top \\ -(\mathbf{Q}\mathbf{V})^\top \end{bmatrix}$. To get $D$ features one simply stacks $n = \frac{D}{2(d+1)}$ such matrices $\mathbf{W}^j = \rho_j \begin{bmatrix} (\mathbf{Q}^j \mathbf{V})^\top \\ -(\mathbf{Q}^j \mathbf{V})^\top \end{bmatrix}$ so that $\mathbf{W} \in \mathbb{R}^{2n(d+1) \times d}$, where only $\mathbf{Q}^j \in \mathbb{R}^{d \times d}$ and $\rho_j$   $(j = 1, \dots, n)$ are generated randomly.

For example, in case of the Gaussian kernel the mapping can be rewritten as [2]

$$\psi_{\mathrm{G}}(\mathbf{x}) = [a_0 \quad a_1 \cos(\mathbf{w}_1^\top \mathbf{x}) \quad \dots \quad a_D \cos(\mathbf{w}_D^\top \mathbf{x}) \quad a_1 \sin(\mathbf{w}_1^\top \mathbf{x}) \quad \dots \quad a_D \sin(\mathbf{w}_D^\top \mathbf{x})].$$

## 5 EXPERIMENTS

We extensively study the proposed method on several established benchmarking datasets: Powerplant, LETTER, USPS, MNIST, CIFAR100 (Krizhevsky & Hinton (2009)), LEUKEMIA (Golub et al. (1999)). In Section 5.2 we show kernel approximation error across different kernels and number of features. We also report the quality of SVM models with approximate kernels on the same data sets in Section 5.3. The compared methods are described below.

### 5.1 METHODS

We present a comparison of our method with estimators based on simple Monte Carlo[3]. The Monte Carlo approach has a variety of ways to generate samples: unstructured Gaussian (Rahimi & Recht (2008)), structured Gaussian (Felix et al. (2016)), random orthogonal matrices (ROM) (Choromanski et al. (2017)).

---

[1]It may be the case when sampling $\rho$ that $1 - \frac{d}{\rho^2} < 0$, simple solution is just to resample $\rho$ to satisfy the non-negativity of the expression.

[2]We do not use reflected points for Gaussian kernel as noted earlier, so $\mathbf{W}^j = \frac{\rho_j}{\sigma}(\mathbf{Q}^j \mathbf{V})^\top$.

[3]We also study quasi-Monte Carlo (Yang et al. (2014)) performance. See Appendix D for details.

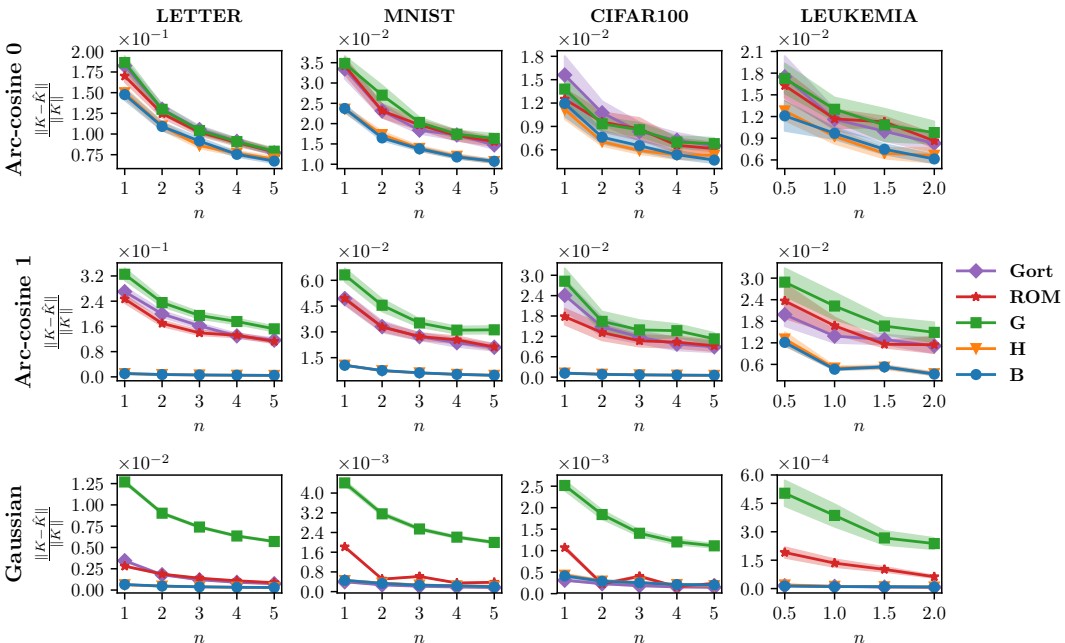

Figure 2: Kernel approximation error across three kernels (columns: arc-cosine 0, arc-cosine 1, Gaussian) on three datasets: LETTER ($d = 16$), MNIST ($d = 784$), CIFAR100 ($d = 3072$) and LEUKEMIA($d = 7129$). Lower is better. The x-axis represents the factor to which we extend the original feature space, $n = \frac{D}{2(d+1)}$, where $d$ is the dimensionality of the original feature space, $D$ is the dimensionality of the new feature space.

**Monte Carlo integration.** The kernel is estimated as $\hat{k}(\mathbf{x}, \mathbf{y}) = \frac{1}{D}\phi(\mathbf{M}\mathbf{x})\phi(\mathbf{M}\mathbf{y})$, where $\mathbf{M} \in \mathbb{R}^{D \times d}$ is a random weight matrix. For unstructured Gaussian based approximation $\mathbf{M} = \mathbf{G}$, where $\mathbf{G}$ is a random matrix with iid $\mathcal{N}(0, 1)$ elements. Structured Gaussian has $\mathbf{M} = \mathbf{G}_{\mathrm{ort}}$, where $\mathbf{G}_{\mathrm{ort}} = \mathbf{D}\mathbf{Q}$, $\mathbf{Q}$ is obtained from RQ decomposition of $\mathbf{G}$ (note that this is not the same $\mathbf{Q}$ used in the quadrature rules), $\mathbf{D}$ is a diagonal matrix with diagonal elements sampled from the $\chi(d)$ distribution. In compliance with the previous work on ROM we use $\mathbf{S}$-Rademacher with three blocks: $\mathbf{M} = \sqrt{d} \prod_{i=1}^{3} \mathbf{S}\mathbf{D}_i$ since three blocks have been shown to yield the best results.

**Quadrature rules.** Our main method that uses stochastic spherical-radial rules with $\mathbf{Q} = \widetilde{\mathbf{B}}$[4] (butterfly matrix) is denoted by $\mathbf{B}$. As mentioned earlier we also include a variant of our algorithm that uses an orthogonal matrix $\mathbf{Q}$ based on a sequence of random reflectors (we denote it as $\mathbf{H}$).

### 5.2 KERNEL APPROXIMATION

To measure kernel approximation quality we use relative error in Frobenius norm $\frac{\|\mathbf{K}-\hat{\mathbf{K}}\|_F}{\|\mathbf{K}\|_F}$, where $\mathbf{K}$ and $\hat{\mathbf{K}}$ denote exact kernel matrix and its approximation. We run experiments for the kernel approximation on a random subset of a dataset (see Appendix D for details). Approximation was constructed for different number of $SR$ samples $n = \frac{D}{2(d+1)}$, where $d$ is an original feature space dimensionality and $D$ is the new one. For the Gaussian kernel we set hyperparameter $\gamma = \frac{1}{2\sigma^2}$ to the same value for all the approximants, while arc-cosine kernels have no hyperparameters.

We run experiments for each [kernel, dataset, $n$] tuple and plot 95% confidence interval around the mean value line. Figure 2 show results for kernel approximation error on LETTER, MNIST, CIFAR100 and LEUKEMIA datasets.

---

[4]$\widetilde{\mathbf{B}} = (\mathbf{B}\mathbf{P})_1(\mathbf{B}\mathbf{P})_2 \ldots (\mathbf{B}\mathbf{P})_3$, where $\mathbf{P}$ is a permutation matrix, for explanation see Appendix C.

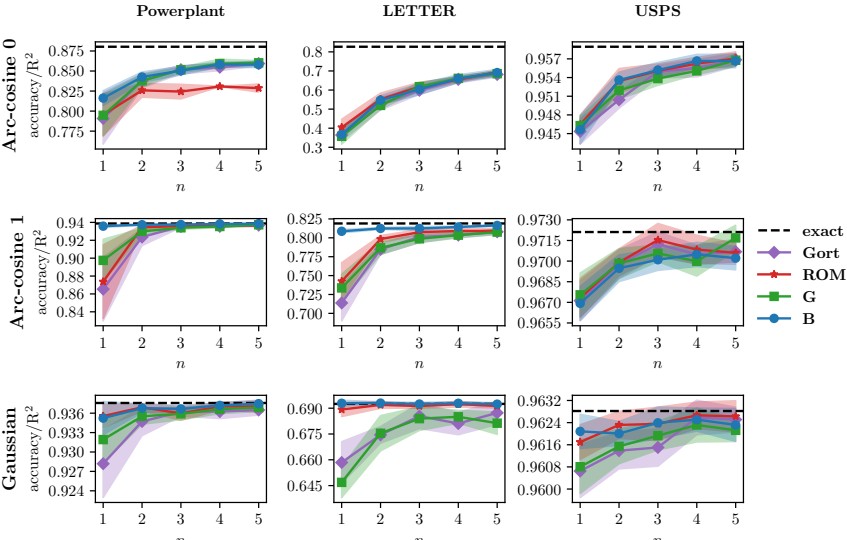

Figure 3: Accuracy/$R^2$ score using embeddings with three kernels (columns: arc-cosine 0, arc-cosine 1, Gaussian) on three datasets (rows: Powerplant, LETTER, USPS). Higher is better. The x-axis represents the factor to which we extend the original feature space, $n = \frac{D}{2(d+1)}$, where $d$ is the dimensionality of the original feature space, $D$ is the dimensionality of the new feature space. We drop one of our methods $\mathbf{H}$ here since its kernel approximation almost coincides with $\mathbf{B}$.

We observe that for the most of the datasets and kernels the methods we propose in the paper ($\mathbf{B}$, $\mathbf{H}$) show better results than the baselines. They do coincide almost everywhere, which is expected, as the $\mathbf{B}$ method is only different from $\mathbf{H}$ in the choice of the matrix $\mathbf{Q}$ to facilitate speed up.

## 5.3 CLASSIFICATION/REGRESSION WITH NEW FEATURES

We report accuracy and $R^2$ scores for the classification and regression tasks on the same data sets (see Figure 3). We examine the performance with the same setting (the number of runs for each [kernel, dataset, $n$] tuple) as in experiments for kernel approximation error, except now we map the whole dataset. We use Support Vector Machines to obtain predictions. We also drop one of our methods $\mathbf{H}$ here since its kernel approximation almost coincides with $\mathbf{B}$.

Kernel approximation error does not fully define the final prediction accuracy – the best performing kernel matrix approximant not necessarily yields the best accuracy or $R^2$ score. However, the empirical results illustrate that our method delivers comparable and often the best quality on the final tasks. We also note that in many cases our method provides greater performance using less number of features $n$, e.g. LETTER and Powerplant datasets with arc-cosine kernel of the first order.

## 6 RELATED WORK

The most popular methods for scaling up kernel methods are based on a low-rank approximation of the kernel using either data-dependent or independent basis functions. The first one includes Nyström method (Drineas & Mahoney (2005)), greedy basis selection techniques (Smola & Schölkopf (2000)), incomplete Cholesky decomposition (Fine & Scheinberg (2001)).

The construction of basis functions in these techniques utilizes the given training set making them more attractive for some problems compared to Random Fourier Features approach. In general, data-dependent approaches perform better than data-independent approaches when there is a gap in the eigen-spectrum of the kernel matrix. The rigorous study of generalization performance of both approaches can be found in (Yang et al. (2012)).

In data-independent techniques, the kernel function is approximated directly. Most of the methods (including the proposed approach) that follow this idea are based on Random Fourier Features

(Rahimi & Recht (2008)). They require so-called weight matrix that can be generated in a number of ways. Le et al. (2013) form the weight matrix as a product of structured matrices. It enables fast computation of matrix-vector products and speeds up generation of random features.

Another work (Felix et al. (2016)) orthogonalizes the features by means of orthogonal weight matrix. This leads to less correlated and more informative features increasing the quality of approximation. They support this result both analytically and empirically. The authors also introduce matrices with some special structure for fast computations. Choromanski et al. (2017) propose a generalization of the ideas from (Le et al. (2013)) and (Felix et al. (2016)), delivering an analytical estimate for the mean squared error (MSE) of approximation.

All these works use simple Monte Carlo sampling. However, the convergence can be improved by changing Monte Carlo sampling to Quasi-Monte Carlo sampling. Following this idea Yang et al. (2014) apply quasi-Monte Carlo to Random Fourier Features. In (Yu et al. (2015)) the authors make attempt to improve quality of the approximation of Random Fourier Features by optimizing sequences conditioning on a given dataset.

Among the recent papers there are works that, similar to our approach, use the numerical integration methods to approximate kernels. While Bach (2017) carefully inspects the connection between random features and quadratures, they did not provide any practically useful explicit mappings for kernels. Leveraging the connection Dao et al. (2017) propose several methods with Gaussian quadratures, among them three schemes are data-independent and one is data-dependent. The authors do not compare them with the approaches for random feature generation other than random Fourier features. The data-dependent scheme optimizes the weights for the quadrature points to yield better performance.

## 7  CONCLUSION

In this work we proposed to apply advanced integration rule that allowed us to achieve higher quality of kernel approximation. Our derivation of the variance of the error implies the dependence of the error on the scale of data, which in case of Gaussian kernel can be interpreted as width of the kernel. However, as we have seen earlier, accuracy on the final task has no direct dependence on the approximation quality, so we can only speculate whether better approximated wide kernels deliver better accuracy compared to the poorer approximated narrow ones. It is interesting to explore this connection in the future work.

To speed up the computations we employed butterfly orthogonal matrices yielding the computational complexity $\mathcal{O}(d \log d)$. Although the procedure we used to generate butterfly matrices claims to produce uniformly random orthogonal matrices, we found that it is not always so. However, the comparison of the method $\mathbf{H}$ (uses properly distributed orthogonal matrices) with method $\mathbf{B}$ (sometimes fails to do so) did not reveal any differences. We also leave it for the future investigation.

Our experimental study confirms that for many kernels on the most datasets the proposed approach delivers better kernel approximation. Additionally, the empirical results showed that the quality of the final task (classification/regression) is also higher than the state-of-the-art baselines. The connection between the final score and the kernel approximation error is to be explored as well.

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

## A  QUADRATURE RULES DETAILS

**The degree of the rules.**  We now discuss the choice of the degree for the $SR$ rule. Genz & Monahan (1999) show that higher degree rules, being more computationally expensive, often bring in only marginal improvement in performance. For these reasons in our experiments we use the rule of degree three $SR^{3,3}$, i.e. a combination of radial rule $R^3$ and spherical rule $S^3$.

## B  VARIANCE OF KERNEL FUNCTION APPROXIMATION USING QUADRATURE RULE

The function $f_{\mathbf{xy}}$ in quadrature rule (7) can be considered as a function of two variables, i.e. $f_{\mathbf{xy}} = \phi(\mathbf{w}^\top\mathbf{x})\phi(\mathbf{w}^\top\mathbf{y}) = g(z_1, z_2)$, where $z_1 = \mathbf{w}^\top\mathbf{x}, z_2 = \mathbf{w}^\top\mathbf{y}$.

In the quadrature rule $\rho \sim \chi(d + 2)$ and $\mathbf{Q}$ is a random orthogonal matrix. Therefore, random variables $\mathbf{w}_i = \mathbf{Q}\mathbf{v}_i$ are uniformly distributed on a unit-sphere.

Now, let's write down 4-th order Taylor expansion with Lagrange remainder of function $g(\rho\mathbf{w}_i^\top\mathbf{x}, \rho\mathbf{w}_i^\top\mathbf{y}) + g(-\rho\mathbf{w}_i^\top\mathbf{x}, -\rho\mathbf{w}_i^\top\mathbf{y})$ around 0 (odd terms cancel out)

$$g(\rho\mathbf{w}_i^\top\mathbf{x}, \rho\mathbf{w}_i^\top\mathbf{y}) + g(-\rho\mathbf{w}_i^\top\mathbf{x}, -\rho\mathbf{w}_i^\top\mathbf{y}) \approx 2g(0,0) + 2\sum_{j=0}^{2}\frac{c_j}{j!(2-j)!}(\mathbf{w}_i^\top\mathbf{x})^j(\mathbf{w}_i^\top\mathbf{y})^{2-j} +$$

$$+ \rho^2 \sum_{j=0}^{4}\frac{d_j^i}{j!(4-j)!}(\mathbf{w}_i^\top\mathbf{x})^j(\mathbf{w}_i^\top\mathbf{y})^{4-j}$$

where $c_j = \frac{\partial^2 g}{\partial z_1^j \partial z_2^{2-j}}(0,0), d_j^i = \frac{\partial^4 g}{\partial z_1^j \partial z_2^{4-j}}(\epsilon_1^i, \epsilon_2^i) + \frac{\partial^4 g}{\partial z_1^j \partial z_2^{4-j}}(\epsilon_3^i, \epsilon_4^i)$, $\epsilon_1^i$ is between 0 and $(\rho\mathbf{w}_i^\top\mathbf{x})$, $\epsilon_2^i$ is between 0 and $(\rho\mathbf{w}_i^\top\mathbf{y})$, $\epsilon_3^i$ is between 0 and $(-\rho\mathbf{w}_i^\top\mathbf{x})$ and $\epsilon_4^i$ is between 0 and $(-\rho\mathbf{w}_i^\top\mathbf{y})$.

Plugging this expression into (5) we obtain

$$SR_{\mathbf{Q},\rho}^{3,3}(f_{\mathbf{xy}}) \approx \left(1 - \frac{d}{\rho^2}\right)g(0,0) +$$

$$+ \frac{d}{2(d+1)\rho^2}\sum_{i=1}^{d+1}\left(2g(0,0) + 2\sum_{j=0}^{2}\frac{c_j}{j!(2-j)!}(\mathbf{w}_i^T\mathbf{x})^j(\mathbf{w}_i^T\mathbf{y})^{2-j} +\right.$$

$$\left. \rho^2 \sum_{j=0}^{4}\frac{d_j}{j!(4-j)!}(\mathbf{w}_i^T\mathbf{x})^j(\mathbf{w}_i^T\mathbf{y})^{4-j}\right) =$$

$$= g(0,0) + \frac{1}{2}\sum_{i=1}^{d+1}S_i,$$

where $S_i = A_i + B_i$, $A_i = \frac{2}{\rho^2}\sum_{j=0}^{2}\frac{c_j}{j!(2-j)!}(\mathbf{w}_i^\top\mathbf{x})^j(\mathbf{w}_i^\top\mathbf{y})^{2-j}$, $B_i = \sum_{j=0}^{4}\frac{d_j^i}{j!(4-j)!}(\mathbf{w}_i^\top\mathbf{x})^j(\mathbf{w}_i^\top\mathbf{y})^{4-j}$, $\mathbf{w}_j = \mathbf{Q}\mathbf{v}_j$, matrix $\mathbf{Q}$ is a random orthogonal matrix uniformly distributed on a set of orthogonal matrices $O(n)$. From uniformity of orthogonal matrix $\mathbf{Q}$ it follows that vector $\mathbf{w}_j$ is uniform on a unit $n$-sphere. Also note that $A_i$ and $B_j$ are independent if $i \neq j$, $B_i$ and $B_j$ are independent if $i \neq j$, however, $A_i$ and $A_j$ are dependent as they have common random variable $\rho$. Therefore, $Cov(S_i, S_j) = Cov(A_i, A_j) = \mathbb{E}(A_i A_j) - (\mathbb{E}(A_i))^2$ if $i \neq j$.

Let us calculate the variance of the estimate.

$$\mathbb{V}\left[SR_{\mathbf{Q},\rho}^{3,3}(f_{\mathbf{xy}})\right] = \frac{1}{4}\mathbb{V}\left[\sum_{j=1}^{d+1}S_j\right] = \frac{1}{4}\sum_{i=1}^{d+1}\mathbb{V}(S_i) + \frac{1}{4}\sum_{i\neq j}^{d+1}Cov(S_i,S_j) \leq$$

$$\leq \frac{1}{4}\sum_{i=1}^{d+1}\mathbb{V}(S_i) + \frac{1}{4}\sum_{i\neq j}^{d+1}\mathbb{E}(A_iA_j) = \frac{1}{4}\sum_{i=1}^{d+1}\mathbb{V}(S_i) +$$

$$+ \mathbb{E}\left(\frac{1}{4\rho^4}\right)\sum_{i\neq j}^{d+1}\left[\mathbb{E}\left(\sum_{k=0}^{2}\frac{c_k}{k!(2-k)!}(\mathbf{w}_i^\top\mathbf{x})^k(\mathbf{w}_i^\top\mathbf{y})^{2-k}\right)\times\right.$$

$$\left.\mathbb{E}\left(\sum_{k=0}^{2}\frac{c_k}{k!(2-k)!}(\mathbf{w}_j^T\mathbf{x})^k(\mathbf{w}_j^T\mathbf{y})^{2-k}\right)\right]. \qquad (9)$$

Distribution of random variable $\rho$ is $\chi(d+2)$, therefore

$$\mathbb{E}\left(\frac{1}{\rho^2}\right) = \frac{1}{d}, \quad \mathbb{E}\left(\frac{1}{\rho^4}\right) = \frac{1}{d(d-2)}, d > 2.$$

1. Now, let us calculate the variance $\mathbb{V}S_i$ from the first term of equation (9)

$$\mathbb{V}S_i = \mathbb{V}(A_i + B_i) = \mathbb{V}(A_i) + \mathbb{V}(B_i) + Cov(A_i,B_i).$$

Let $\|\mathbf{x}\| \geq \|\mathbf{y}\|$. Then

$$\mathbb{V}(B_i) = \mathbb{V}\left(\sum_{j=0}^{4}\frac{d_j^i(\mathbf{w}_i^\top x)^j(\mathbf{w}_i^\top\mathbf{y})^{4-j}}{j!(4-j)!}\right) \leq \mathbb{E}\left(\sum_{j=0}^{4}\frac{d_j^i(\mathbf{w}_i^\top\mathbf{x})^j(\mathbf{w}_i^\top\mathbf{y})^{4-j}}{j!(4-j)!}\right)^2 =$$

$$= \mathbb{E}\left(\frac{(d_0^i)^2(\mathbf{w}^\top\mathbf{y})^8 + (d_4^i)^2(\mathbf{w}^\top\mathbf{x})^8}{596} + \frac{(4(d_1^i)^2 + 3d_0^id_2^i)(\mathbf{w}^\top\mathbf{x})^2(\mathbf{w}^\top\mathbf{y})^6}{144} + \right.$$

$$\frac{(4(d_3^i)^2 + 3d_4^id_2^i)(\mathbf{w}^\top\mathbf{x})^6(\mathbf{w}^\top\mathbf{y})^2}{144} +$$

$$\left.\frac{(1296(d_2^i)^2 + 72d_0^id_4^i + 1152d_1^id_3^i)(\mathbf{w}^\top\mathbf{x})^4(\mathbf{w}^\top\mathbf{y})^4}{20736}\right) \leq$$

$$\leq \frac{M_1^2}{298}\mathbb{E}\left(\mathbf{w}^T\mathbf{x}\right)^8 + \frac{7M_1^2}{72}\|\mathbf{x}\|^2\mathbb{E}\left(\mathbf{w}^T\mathbf{x}\right)^6 + 0.122M_1^2\|\mathbf{x}\|^4\mathbb{E}\left(\mathbf{w}^T\mathbf{x}\right)^4, \qquad (10)$$

where $M_1 = \max\left\{\sup_z\left|\frac{\partial^4 g}{\partial z_1^4}(z_1,z_2)\right|, \sup_z\left|\frac{\partial^4 g}{\partial z_2^4}(z_1,z_2)\right|, \sup_z\left|\frac{\partial^4 g}{\partial z_1^2\partial z_2^2}(z_1,z_2)\right|\right\}$.

To calculate expectations of the form $\mathbb{E}(\mathbf{w}^\top x)^k$ we will use expression for integral of monomial over unit sphere Baker (1997)

$$J(k_1,k_2,\ldots,k_d) = \int_{S_{d-1}}\mathbf{x}_1^{k_1}\mathbf{x}_2^{k_2}\cdots\mathbf{x}_d^{k_d}d\mathbf{x} = \sigma_d\frac{(k_1-1)!!(k_2-1)!!\cdots(k_d-1)!!}{d(d+2)\cdots(d+|k|-2)}, \qquad (11)$$

where $k_i = 2s_i, s_i \in \mathbb{Z}_+, |k| = \sum_i k_i$, $\sigma_d$ is a volume of an $d$-dimensional unit sphere.

For example, let us show how to calculate $\mathbb{E}(\mathbf{w}^\top x)^4$:

$$\mathbb{E}(\mathbf{w}^\top\mathbf{x})^4 = \mathbb{E}\sum_{i,j,k,m}\mathbf{w}_i\mathbf{w}_j\mathbf{w}_k\mathbf{w}_m\mathbf{x}_i\mathbf{x}_j\mathbf{x}_k\mathbf{x}_m =$$

$$= \left[\text{thanks to symmetry all terms, for which at least one index doesn't coincide with}\right.$$

$$\left.\text{other indices, are equal to 0.}\right] = \mathbb{E}\left[\sum_i\mathbf{w}_i^4\mathbf{x}_i^4 + 3\sum_{i\neq j}\mathbf{w}_i^2\mathbf{w}_j^2\mathbf{x}_i^2\mathbf{x}_j^2\right] =$$

$$= \frac{3}{d(d+2)}\sum_i\mathbf{x}_i^4 + \frac{3}{d(d+2)}\sum_{i\neq j}\mathbf{x}_i^2\mathbf{x}_j^2 = \frac{3}{d(d+2)}\|\mathbf{x}\|^4.$$

Using the same technique for (10) we obtain

$$\mathbb{V}(B_i) \leq M_1^2 \|\mathbf{x}\|^8 \left( \frac{8.46}{d(d+2)(d+4)(d+6)} + \frac{10.21}{d(d+2)(d+4)} + \frac{0.37}{d(d+2)} \right) \leq$$

$$\leq \frac{2.66 M_1^2 \|\mathbf{x}\|^8}{d(d+2)} \tag{12}$$

For the variance of $A_i$ we have

$$\mathbb{V}(A_i) = \mathbb{V}\left( \frac{2}{\rho^2} \sum_{j=0}^{2} \frac{c_j (\mathbf{w}_i^T \mathbf{x})^j (\mathbf{w}_i^T \mathbf{y})^{2-j}}{j!(2-j)!} \right) \leq \mathbb{E}\left( \frac{2}{\rho^2} \sum_{j=0}^{2} \frac{c_j (\mathbf{w}_i^T \mathbf{x})^j (\mathbf{w}_i^T \mathbf{y})^{2-j}}{j!(2-j)!} \right)^2 =$$

$$= \frac{4}{d(d-2)} \mathbb{E}\left( \sum_{j=0}^{2} \frac{c_j (\mathbf{w}_i^T \mathbf{x})^j (\mathbf{w}_i^T \mathbf{y})^{2-j}}{j!(2-j)!} \right)^2 =$$

$$= \frac{4}{d(d-2)} \left( \frac{3c_0^2 \|\mathbf{x}\|^4 + 3c_2^2 \|\mathbf{y}\|^4 + (4c_1^2 + 2c_0 c_2)(\|\mathbf{x}\|^2 \|\mathbf{y}\|^2 + 2(\mathbf{x}^T \mathbf{y})^2)}{4d(d+2)} \right) \leq$$

$$\leq \left[ \text{we assume that } \|\mathbf{x}\| \geq \|\mathbf{y}\| \right] \leq \frac{24}{d^2(d^2-4)} M_2^2 \|\mathbf{x}\|^4, \tag{13}$$

where $M_2 = \max\limits_{j=0,1,2} \left\{ \left| \frac{\partial^2 g}{\partial z_1^j \partial z_2^{2-j}}(0,0) \right| \right\}$.

Let's estimate covariance $Cov(A_i, B_i)$:

$$Cov(A_i, B_i) \leq \mathbb{E}(A_i B_i) + |\mathbb{E}A_i \mathbb{E}B_i|.$$

The first term of the right hand side:

$$\mathbb{E}\left( \frac{2}{\rho^2} \sum_{j=0}^{2} \frac{c_j (\mathbf{w}_i^T \mathbf{x})^j (\mathbf{w}_i^T \mathbf{y})^{2-j}}{j!(2-j)!} \sum_{j=0}^{4} \frac{d_j^i (\mathbf{w}_i^T \mathbf{x})^j (\mathbf{w}_i^T \mathbf{y})^{4-j}}{j!(4-j)!} \right) \leq$$

$$\left[ \text{we assume that } \|\mathbf{x}\| \geq \|\mathbf{y}\| \right] \leq \frac{210 M_1 M_2 \|\mathbf{x}\|^6}{d^2(d+2)(d+4)}.$$

The second term $\mathbb{E}A_i \mathbb{E}B_i$ (again for $\|\mathbf{x}\| \geq \|\mathbf{y}\|$)

$$|\mathbb{E}A_i \mathbb{E}B_i| \leq \frac{2 M_1 M_2 \|\mathbf{x}\|^6}{d^3(d+2)}.$$

Combining the derived inequalities we obtain

$$\mathbb{V}S_i \leq \frac{24}{d^2(d^2-4)} M_2^2 \|\mathbf{x}\|^4 + \frac{2.66 M_1^2 \|\mathbf{x}\|^8}{d(d+2)} + \frac{210 M_1 M_2 \|\mathbf{x}\|^6}{d^2(d+2)(d+4)} + \frac{2 M_1 M_2 \|\mathbf{x}\|^6}{d^3(d+2)} \leq$$

$$\leq \frac{2.66 M_1^2 \|\mathbf{x}\|^8}{d(d+2)} + \frac{212 M_1 M_2 \|\mathbf{x}\|^6}{d^3(d+2)} + \frac{24 M_2^2 \|\mathbf{x}\|^4}{d^2(d^2-4)}. \tag{14}$$

2. Now, let's examine the expectation of the second term in (9):

$$\mathbb{E}\left( \sum_{k=0}^{2} \frac{c_k}{k!(2-k)!} (\mathbf{w}_i^\top \mathbf{x})^k (\mathbf{w}_i^\top \mathbf{y})^{2-k} \right) =$$

$$\frac{c_0}{2} \mathbb{E}(\mathbf{w}_i^\top \mathbf{y})^2 + c_1 \mathbb{E}(\mathbf{w}_i^\top \mathbf{x})(\mathbf{w}_i^\top \mathbf{y}) + \frac{c_2}{2} \mathbb{E}(\mathbf{w}_i^\top \mathbf{x})^2 \leq$$

$$\leq \left[ \text{we assume that } \|\mathbf{x}\| \geq \|\mathbf{y}\| \right] \leq \frac{M_2 \|\mathbf{x}\|^2}{d}. \tag{15}$$

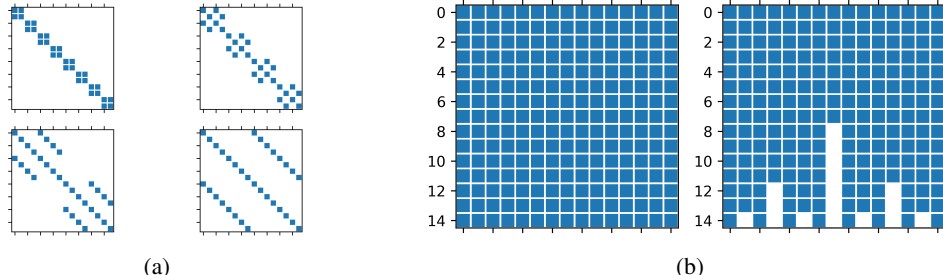

Figure 4: (a) Butterfly orthogonal matrix factors for $d = 16$. (b) Sparsity pattern for $\mathbf{BPBPBP}$ (left) and $\mathbf{B}$ (right), $d = 15$.

Substituting (14) and (15) into (9) we obtain

$$\mathbb{V}\left[ SR^{3,3}_{\mathbf{Q},\rho}(f_{\mathbf{xy}}) \right] \le (d+1)\left( \frac{2.66 M_1^2 \|\mathbf{x}\|^8}{d(d+2)} + \frac{212 M_1 M_2 \|\mathbf{x}\|^6}{d^3(d+2)} + \frac{24 M_2^2 \|\mathbf{x}\|^4}{d^2(d^2-4)} \right) +$$

$$+ \frac{1}{4d(d-2)} d(d-1) \left( \frac{M_2 \|\mathbf{x}\|^2}{d} \right)^2 \le$$

$$\le \frac{2.66 M_1^2 \|\mathbf{x}\|^8}{d} + \frac{212 M_1 M_2 \|\mathbf{x}\|^6}{d^3} + \frac{(d+95) M_2^2 \|\mathbf{x}\|^4}{4d^2(d-2)}.$$

And finally

$$\mathbb{V}\left[ \frac{1}{n} \sum_{i=1}^{n} SR^{3,3}_{\mathbf{Q}_i,\rho_i}(f_{\mathbf{xy}}) \right] \le \frac{2.66 M_1^2 \|\mathbf{x}\|^8}{nd} + \frac{212 M_1 M_2 \|\mathbf{x}\|^6}{nd^3} + \frac{(d+95) M_2^2 \|\mathbf{x}\|^4}{4nd^2(d-2)}.$$

## C  BUTTERFLY MATRICES GENERATION DETAILS

### C.1  NOT A POWER OF TWO

We discuss here the procedure to generate butterfly matrices of size $d \times d$ when $d$ is not a power of 2.

Let the number of butterfly factors $k = \lceil \log d \rceil$. Then $\mathbf{B}^{(d)}$ is constructed as a product of $k$ factor matrices of size $d \times d$ obtained from $k$ matrices used for generating $\mathbf{B}^{(2^k)}$. For each matrix in the product for $\mathbf{B}^{(2^k)}$, we delete the last $2^k - d$ rows and columns. We then replace with 1 every $c_i$ in the remaining $d \times d$ matrix that is in the same column as deleted $s_i$.

For the cases when $d$ is not a power of two, the resulting $\mathbf{B}$ has deficient columns with zeros (Figure 4b, right), which introduces a bias to the integral estimate. To correct for this bias one may apply additional randomization by using a product $\mathbf{BP}$, where $\mathbf{P} \in \{0,1\}^{d \times d}$ is a permutation matrix. Even better, use a product of several $\mathbf{BP}$'s: $\widetilde{\mathbf{B}} = (\mathbf{BP})_1 (\mathbf{BP})_2 \dots (\mathbf{BP})_t$. We set $t = 3$ in the experiments.

### C.2  BUTTERFLY RANDOMIZATION

The key to uniformly random orthogonal butterfly matrix $\mathbf{B}$ is the sequence of $d-1$ angles $\theta_i$. To get $\mathbf{B}^{(d)}$ Haar distributed, we follow Fang & Li (1997) algorithm that first computes a uniform random point $\mathbf{u}$ from $U_d$. It then calculates the angles by taking the ratios of the appropriate $\mathbf{u}$ coordinates $\theta_i = \frac{u_i}{u_{i+1}}$, followed by computing cosines and sines of the $\theta$'s.

## D  KERNEL APPROXIMATION ERRORS ON DIFFERENT DATA SETS

Here we discuss the datasets that did not appear in the main body of the paper. Table 2 displays the settings for the experiments across the datasets. Figure 5 shows the results for the kernel approxi-

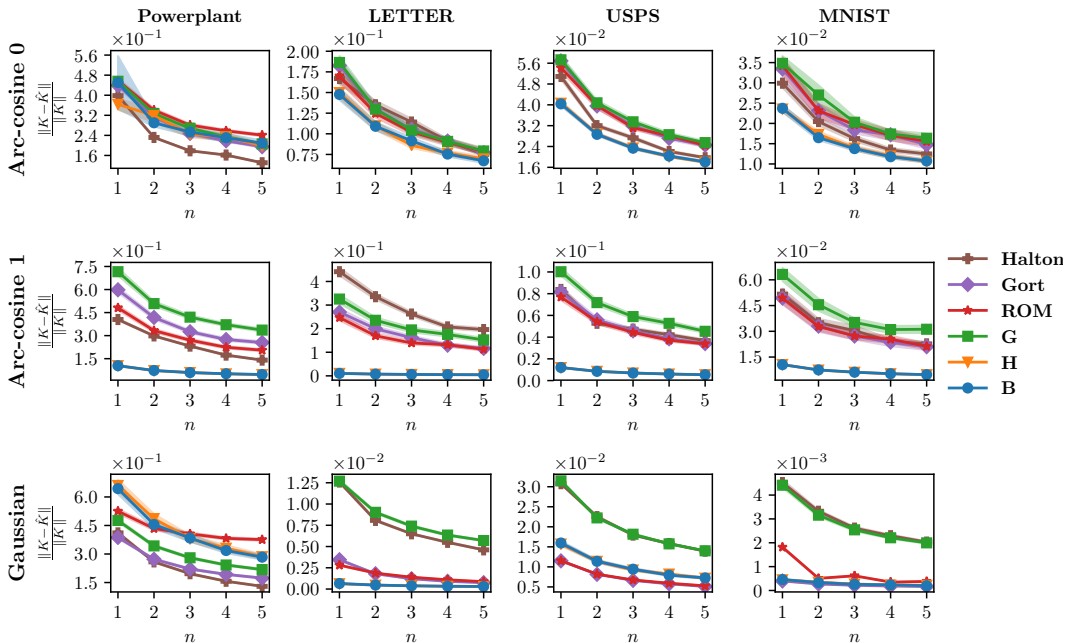

Figure 5: Kernel approximation error across three kernels (columns: arc-cosine 0, arc-cosine 1, Gaussian) on three datasets: Powerplant ($d = 4$), LETTER ($d = 16$), USPS ($d = 256$), MNIST ($d = 784$). Lower is better. The x-axis represents the factor to which we extend the original feature space, $n = \frac{D}{2(d+1)}$, where $d$ is the dimensionality of the original feature space, $D$ is the dimensionality of the new feature space.

mation error on Powerplant, LETTER, USPS and MNIST datasets. For these datasets we include QMC with Halton sequences into comparison as well.

Table 2: Experimental settings for the datasets. $N$ is the total number of objects, $d$ is dimensionality of the original feature space.

| Dataset | $N$ | $d$ | Number of samples | Number of runs |
|---|---|---|---|---|
| Powerplant | 9568 | 4 | 550 | 500 |
| LETTER | 20000 | 16 | 550 | 500 |
| USPS | 9298 | 256 | 550 | 500 |
| MNIST | 70000 | 784 | 50 | 50 |
| CIFAR100 | 60000 | 3072 | 50 | 50 |
| LEUKEMIA | 72 | 7129 | 10 | 10 |

Quasi-Monte Carlo integration boasts improved rate of convergence $1/D$ compared to $1/\sqrt{D}$ of Monte Carlo, however, empirical results illustrate its performance is poorer than that of orthogonal random features (Felix et al. (2016)). It also has larger constant factor hidden under $\mathcal{O}$ notation and higher complexity. For QMC the weight matrix $\mathbf{M}$ is generated as a transformation of quasi-random sequences. We run our experiments with Halton sequences in compliance with the previous work (see Figure 5 with QMC method included).

Although, for the arc-cosine kernels, our methods are the best performing estimators, for the Gaussian kernel the error is not always the lowest one and depends on the dataset, e.g. on the USPS dataset the lowest is Monte Carlo with ROM. However, for the most of the datasets we demonstrate superiority of our approach with this kernel.

We also notice that the dataset with a small amount of features, Powerplant, enjoys Halton and Orthogonal Random Features best, while ROM's convergence stagnates at some point. This could be due the small input feature space with $d = 4$ and we leave it for the future investigation.

# E    COMPARISON WITH GAUSSIAN QUADRATURES

We also included subsampled dense grid method from Dao et al. (2017) into our comparison as it is the only data-independent approach from the paper that is shown to work well. We reimplemented code for the paper to the best of our knowledge since it is not open sourced. We run comparison on all datasets with Gaussian kernel. The Figure 6 illustrates the performance on the LETTER dataset across different expansions. We can see almost coinciding performance of the method (denoted **GQ**) with the baseline RFF (denoted **G**). For other datasets the figures are very similar to the case of LETTER, with RFF and **GQ** methods showing nearly matching relative error of kernel approximation.

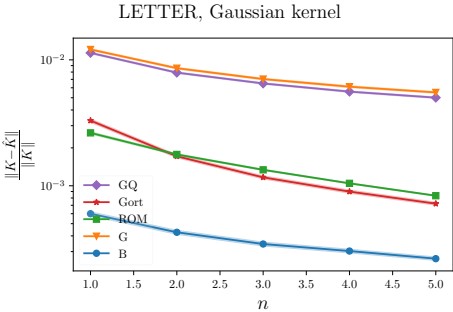

Figure 6: Kernel approximation error for Gaussian kernel on LETTER dataset. Subsampled dense grid method (denoted **GQ**) from Dao et al. (2017) show very similar performance to the baseline random Fourier features (denoted **G** on the picture).

It should also be noted that explicit mappings produced by Gaussian quadratures do not possess any convenient structure and, thus, cannot boast any better computational complexity.

# F    WALLTIME EXPERIMENT

We have also run the time measurement on our somewhat unoptimized implementation of the proposed method. Indeed, Figure 7 demonstrates that the method scales as theoretically predicted with larger dimensions thanks to the structured nature of the mapping.

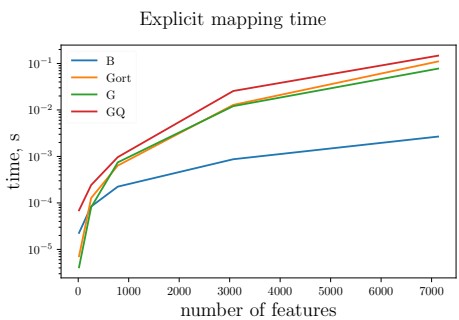

Figure 7: Walltime experiment measures the time spent on applying the explicit mapping across different methods. The x-axis represents the 5 datasets with increasing input number of features: LETTER ($d = 16$), USPS ($d = 256$), MNIST ($d = 784$), CIFAR100 ($d = 3072$) and LEUKEMIA ($d = 7129$). Thanks to structured explicit mapping of the proposed method **B**, it is favorable for higher dimensional data.

