# OpenReview forum: "Quadrature-based features for kernel approximation"
_ICLR.cc/2018/Conference — Reject_

### Official Review · AnonReviewer3 · 2017-11-26
**Interesting paper, but lacks novelty and comparison to existing work**

**Rating:** 4
**Confidence:** 3

**Review:**

The paper proposes to improve the kernel approximation of random features by using quadratures, in particular, stochastic spherical-radial rules. The quadrature rules have smaller variance given the same number of random features, and experiments show its reconstruction error and classification accuracies are better than existing algorithms.

It is an interesting paper, but it seems the authors are not aware of some existing works [1, 2] on quadrature for random features. Given these previous works, the contribution and novelty of the paper is limited.

[1] Francis Bach. On the Equivalence between Kernel Quadrature Rules and Random Feature Expansions. JMLR, 2017.
[2] Tri Dao, Christopher De Sa, Christopher Ré. Gaussian Quadrature for Kernel Features. NIPS 2017

---

> ### Author Response · Authors · 2018-01-04
> **Thanks for the review**
>
> We thank the reviewer for directing us to alternative work on quadrature for random features. We have updated the related work section to include both papers. We also address the differences in what follows.
>
> [1] In this paper the authors consider a problem of calculating integral of a function with respect to some probability measure. They show that this problem is equivalent to the random feature problem for a specific feature map. It is known that if we sample weights from the distribution, the rate of convergence is square root of (1 / n). In this paper an optimal distribution for weights is derived, which leads to better rate of convergence. Actually, the distribution depends on the so called leverage score. However, they operate in Hilbert space and calculation of leverage score requires inversion of integral operator. Such integral operators are infinite-dimensional, so it is hard to compute them in practice. To make their algorithm practical, they propose the following procedure: sample large amount (N) of weights W from the initial distribution; then calculate the density of optimized distribution on the set of sampled weights; after that resample small amount of weights from generated weights W. The complexity of the algorithm is O(N^3).
> The main objective of the paper is to numerically calculate the integral of a function. The problem is equivalent to generation of random features for kernel approximation, but they don’t study explicitly the kernel approximation. So, in the experimental section they do not check the error of approximation of kernel function. They just take several functions and calculate quadrature errors on them. Thus, the paper does not really provide any practically useful explicit mappings for kernel approximation.
>
> [2] The authors propose several methods, among them three schemes are data-independent and one is data-dependent. We cannot directly compare our method with data-dependent one because our method does not use the data to construct mappings, i.e. is data-independent (a brief discussion on the difference of data-dependent and independent techniques can be found in the related section work). However, we note that one can apply the proposed data-dependent scheme to our method to learn the weights of the points as well. As a matter of fact, one can use random points and learn the weights for them in the proposed fashion. Thus, we only consider the data-independent approaches.
> Dense grid and sparse grid methods are shown to be problematic in the paper. Dense grid is known to suffer heavily from the curse of dimensionality, while sparse grid yields high error rate. The last data-independent approach is subsampling dense grid according to the distribution on the weights of the points. Unfortunately, the code for the paper is not yet available, but we have reimplemented it to the best of our knowledge and ran experiments to compare with the proposed data-independent subsampled dense grid approach. We tested the subsampled approach on all datasets with Gaussian kernel and, unfortunately, it showed nearly the same performance as random Fourier features (RFF), which was indeed shown in the paper for the ANOVA kernel as well. We added the figure with the comparison to the Appendix section E.
>
> To sum things up,
> 1) the first paper does not provide practically useful explicit mappings for kernel approximation (due to the complexity O(N^3), where N is the number of features), while
>
> 2) the second paper has one data-independent method that is eligible for the comparison. The subsampled dense grid method from [2] showed higher kernel approximation error than our method  across all the datasets.
>
> 3) We updated the text of the paper to reflect this comparison. We also included a brief discussion of both papers to the related work section.

---

### Official Review · AnonReviewer2 · 2017-11-27
**incremental development in random feature map approach**

**Rating:** 7
**Confidence:** 5

**Review:**

The authors offer a novel version of the random feature map approach to approximately solving large-scale kernel problems: each feature map evaluates the "fourier feature" corresponding to the kernel at a set of randomly sampled quadrature points. This gives an unbiased kernel estimator; they prove a bound its variance and provide experiment evidence that for Gaussian and arc-cos kernels, their suggested qaudrature rule outperforms previous random feature maps in terms of kernel approximation error and in terms of downstream classification and regression tasks. The idea is straightforward, the analysis seems correct, and the experiments suggest the method has superior accuracy compared to prior RFMs for shift-invariant kernels. The work is original, but I would say incremental, and the relevant literature is cited.

The method seems to give significantly lower kernel approximation errors, but the significance of the performance difference in downstream ML tasks is unclear --- the confidence intervals of the different methods overlap sufficiently to make it questionable whether the relative complexity of this method is worth the effort. Since good performance on downstream tasks is the crucial feature that we want RFMs to have, it is not clear that this method represents a true improvement over the state-of-the-art. The exposition of the quadrature method is difficult to follow, and the connection between the quadrature rules and the random feature map is never explicitly stated: e.g. equation 6 says how the kernel function is approximated as an integral, but does not give the feature map that an ML practitioner should use to get that approximate integral.

It would have been a good idea to include figures showing the time-accuracy tradeoff of the various methods, which is more important in large-scale ML applications than the kernel approximation error. It is not clear that the method is *not* more expensive in practice than previous methods (Table 1 gives superior asymptotic runtimes, but I would like to see actual run times, as evaluating the feature maps sound relatively complicated compared to other RFMs). On a related note, I would also like to have seen this method applied to kernels where the probability density in the Bochner integral was not the Gaussian density (e.g., the Laplacian kernel): the authors suggested that their method works there as well when one uses a Gaussian approximation of the density (which is not clear to me),  --- and it may be the case that sampling from their quadrature distribution is faster than sampling from the original non-Gaussian density.

---

> ### Author Response · Authors · 2018-01-04
> **Thanks for the review**
>
> We deeply appreciate the reviewer for a constructive and comprehensive feedback.
>
> About the performance in downstream ML tasks and explicit mapping:
> We have updated the paper to include the explicit map construction (Section 4.3), which has little additional complexity compared to state-of-the-art Monte Carlo methods, such as Random Orthogonal Features. While our method similar to MC methods only needs matrix multiplication to produce random features, it provides empirically better kernel approximation and in many instances better downstream quality.
>
> Walltime experiments:
> We have updated the Appendix of the paper to include the actual runtimes, which show that for higher dimensions there is indeed an advantage. The figure we added shows that our somewhat unoptimized implementation of the proposed method (B) indeed scales as theoretically predicted with larger dimensions thanks to the structured nature of the mapping.
>
> Kernels with other densities:
> We are deeply sorry and have removed this unsupported claim from the text of the paper. While the approximation is implementable for Laplacian kernel, unfortunately, we found it to be not accurate enough, one would need to use other quadratures to approximate kernels with different densities.

---

### Official Review · AnonReviewer1 · 2017-11-28
**Interesting method with good empirical result, but not enough insights on why**

**Rating:** 6
**Confidence:** 4

**Review:**

This paper shows that techniques due to Genz & Monahan (1998) can be used to achieve low kernel approximation error under the framework of random fourier feature.

Pros

1. It is new to apply quadrature rules to improve kernel approximation. The only other work I found is
Gaussian Quadrature for Kernel Features NIPS 2017.
The work is pretty recent so the author might not know it when submitting the paper. But in either case, it will be good to discuss the connections.

2. The proposed method is shown to outperform a few baselines empirically.

Cons

1. I don’t find the theoretical analysis to be very useful. In particular, the theorem shows that the kernel approximation error is O(1/D), which is the same as the original RFF paper. Unless the paper can provide a better characterization of the constants (like the ORF paper), it does not provide much insight in the proposed method. Unlike deep neural networks, since RFF is such a simple model, I think providing precise theoretical understanding is crucial.

2. Approximating an integral is a well-studied topic. I do not find a good discussion on all the possible methods. Why is Genz & Monahan 1998 better than other alternatives such as Monte-Carlo, QMC etc? One argument seems to be “for kernels with specific specific integrand one can improve on its properties”. But this trick can be used for Monte-Carlo as well. And I do not see benefit of this trick in the curves.

3. When choosing the orthogonal matrix, I think one obvious choice is to sample a matrix from the Stiefel manifold (the Q matrix of a random Gaussian). This baseline should be added in additional to H and B.

4. A wall-time experiment is needed to justify the speedup.

Minor comments:
“For kennels with q(w) other than Gaussian… obtain very accurate results with little effort by using Gaussian approximation of q(w)”. What is the citation of this in the kernel approximation context?

---

> ### Author Response · Authors · 2018-01-04
> **Thanks for the review**
>
> We thank the reviewer for the helpful and thorough feedback.
>
> About Gaussian Quadrature for Kernel Features NIPS 2017:
> Thank you for pointing this paper. The authors propose several methods, among them three schemes are data-independent and one is data-dependent. We cannot directly compare our method with data-dependent one because our method does not use any data to construct mappings, i.e. is data-independent (a brief discussion on the difference of data-dependent and independent techniques can be found in the related section work). However, we note that one can apply the proposed data-dependent scheme to our method to learn the weights of the points as well. As a matter of fact, one can use random points and learn the weights for them in the proposed fashion. Thus, we only consider the data-independent approaches.
> Dense grid and sparse grid methods are shown to be problematic in the paper. Dense grid is known to suffer heavily from the curse of dimensionality, while sparse grid yields high error rate. The last data-independent approach is subsampling dense grid according to the distribution on the weights of the points. Unfortunately, the code for the paper is not yet available, but we have reimplemented it to the best of our knowledge and ran experiments to compare with the proposed data-independent subsampled dense grid approach. We tested the subsampled approach on all datasets with Gaussian kernel and, unfortunately, it showed almost the same performance as random Fourier features (RFF), which has been shown in the paper for the ANOVA kernel as well. We added the figure with the comparison to the Appendix section E.
>
> In a nutshell,
> 1) only subsampled dense grid method was found eligible for the comparison,
>
> 2) it showed higher kernel approximation error than our method across all the datasets.
>
> 3) We updated the text of the paper to reflect this comparison and added a brief discussion of the paper to the related work section.
>
> Regarding theoretical analysis:
> Indeed, we did not elaborate much on the convergence, leaving it to show the similar rates as all state-of-the-art MC methods. However, to the best of our knowledge we were the first to highlight the dependence of the kernel approximation quality on the scale of the data.
>
> About MC and QMC methods for approximating an integral:
> Indeed, we did not show theoretically that MC or QMC methods are worse than the one we propose, however we conducted an extensive study that showed that on most of the datasets the proposed quadrature based features approximate the kernel with lower relative error. Theoretical proof remains an open question for further work.
> Another point about QMC is something that has already been raised in the previous literature (we also noted this in Appendix section D), although QMC provides better asymptotic convergence than MC, it has larger constant factors hidden under O notation and has higher computational complexity along with lower empirical performance.
>
> About the other option for an orthogonal matrix,
> Thank you for the proposed option. We have tried it while preparing the paper, it did show similar/equivalent performance to the ones we used, though it is not sparse or structured as butterflies and, thus, has no computational advantage. That was the reason we did not include it into the paper.
>
> As for the last pointed out con in the paper,
> we have updated the text to include the walltime experiment in the Appendix section F. The figure we added shows that our even somewhat unoptimized implementation of the proposed method (B) indeed scales as theoretically predicted with larger dimensions thanks to the structured nature of the mapping.
>
> Addressing the minor comment:
> We apologize for this unsupported claim and have removed it from the text of the paper. Although the Laplacian kernel approximation can be implemented, unfortunately, it would not be accurate enough and one would need to use other quadratures to approximate kernels with densities other than Gaussian.

---

### Decision · Program_Chairs · 2018-01-29
**ICLR 2018 Conference Acceptance Decision**

**Decision:**

Reject

**Comment:**

This an interesting new contribution to construction of random features for approximating kernel functions. While the empirical results look promising, the reviewers have raised concerns about not having insights into why the approach is more effective;  the exposition of the quadrature method is difficult to follow; and the connection between the quadrature rules and the random feature map is never explicitly stated. Some comparisons are missing (e.g., QMC methods). As such the paper will benefit from a revision and is not ready for ICLR-2018 acceptance.